# Simultaneous Measurement of Temperature and Refractive Index Using High Temperature Resistant Pure Quartz Grating Based on Femtosecond Laser and HF Etching

**DOI:** 10.3390/ma14041028

**Published:** 2021-02-22

**Authors:** Na Zhao, Qijing Lin, Kun Yao, Fuzheng Zhang, Bian Tian, Feng Chen, Zhuangde Jiang

**Affiliations:** 1State Key Laboratory for Manufacturing Systems Engineering, Xi’an Jiaotong University, Xi’an 710049, China; zn2015@stu.xjtu.edu.cn (N.Z.); vinsent@stu.xjtu.edu.cn (K.Y.); xjzfz123@stu.xjtu.edu.cn (F.Z.); t.b12@mail.xjtu.edu.cn (B.T.); chenfeng@mail.xjtu.edu.cn (F.C.); zdjiang@mail.xjtu.edu.cn (Z.J.); 2Collaborative Innovation Center of High-End Manufacturing Equipment, Xi’an Jiaotong University, Xi’an 710054, China

**Keywords:** hollow needle, temperature sensing, refractive index sensing, optical fiber sensor, fiber Bragg grating (FBG), femtosecond laser, hydrofluoric acid

## Abstract

The optical fiber temperature and refractive index sensor combined with the hollow needle structure for medical treatment can promote the standardization of traditional acupuncture techniques and improve the accuracy of body fluid analysis. A double-parameter sensor based on fiber Bragg grating (FBG) is developed in this paper. The sensor materials are selected through X-ray diffraction (XRD) analysis, and the sensor sensing principle is theoretically analyzed and simulated. Through femtosecond laser writing pure silica fiber, a high temperature resistant wavelength type FBG temperature sensor is obtained, and the FBG is corroded by hydrofluoric acid (HF) to realize a high-sensitivity intensity-type refractive index sensor. Because the light has dual characteristics of energy and wavelength, the sensor can realize simultaneous dual-parameter sensing. The light from the lead-in optical fiber is transmitted to the sensor and affected by temperature and refractive-index; then, the reflection peak is reflected back to the lead-out fiber by the FBG. The high temperature response and the refractive index response of the sensor were measured in the laboratory, and the high temperature characteristics of the sensor were verified in the accredited institute. It is demonstrated that the proposed sensor can achieve temperature sensing up to 1150 °C with the sensitivity of 0.0134 nm/°C, and refractive sensing over a refractive range of 1.333 to 1.4027 with the sensitivity of −49.044 dBm/RIU. The sensor features the advantages of two-parameter measurement, compact structure, and wide temperature range, and it exhibits great potential in acupuncture treatment.

## 1. Introduction

Fire needle therapy has a history of thousands of years in China, and the earliest recorded classic is “Huangdi Neijing”. With the clinical application and research of doctors in the past dynasties, fire needle therapy has been continuously developed and improved, and the hollow fire needle [1,2,3] has become a commonly used acupuncture needle because of its fine internal structure. The hollow fire needle [4,5,6,7,8] has the functions of warming meridians and dredging collaterals, dispelling wind and cold, which can be applied in the treatment of arthralgia, gastroptosis, malnutrition in children, irregular menstruation, dysmenorrhea, diarrhea, dysentery, rubella, etc. Burning needles is a key step in using fire needles. The “Acupuncture Dacheng · Fire Needle” [9] says that “Burning on the lamp will make it bright red, and the prescription will be effective. If it is not red, it will not cure the disease and harm people.” Therefore, the needle must be heated before use. The needle burning process uses an alcohol lamp, and its heating temperature can reach above 400 to 1000 °C. In addition, in the process of needle application, because warm needle moxibustion has a special effect on rheumatism and other cold diseases, it is also a research hotspot. In the process of retaining the needle, the moxa is wrapped around the needle handle and ignited. The needle body transfers heat into the acupuncture points. After the fire needle enters the body, it is difficult to maintain a stable temperature in the body. Therefore, it is necessary to design a temperature sensor to be placed in the hollow needle to monitor the needle retention process so that the temperature can be adjusted by moxa outside the body to achieve better curative effect. It is worth noting that the flame temperature is close to 1000 °C in the needle burning stage, so the sensor needs to have high temperature resistance characteristics in order to achieve effective temperature monitoring of the fire needle.

The sensor placed in the hollow fire needle has many requirements, such as good insulation, low cost, light weight, stable chemical property, compact structure. Compared with electrical sensors, optical fiber sensors [10,11,12,13,14,15,16,17,18,19] have stable chemical properties, high temperature resistance and high sensitivity. It is a research hotspot in biomedical temperature and refractive index measurement. The structure classification includes Fabry–Perot interferometer [10,11], Michelson interferometer [12], fiber Bragg grating (FBG), Mach–Zehnder interferometer [13,14,15], etc. Among them, the FBG [16,17,18,19,20,21,22,23,24,25] has a higher maturity and can be widely used in detection and monitoring sensing systems. In 1978, Hill K.O et al. [20] discovered the photosensitive effect of the fiber in germanium-doped silica fiber for the first time and used the standing wave writing method to make the world’s first FBG. In 1995, L. Dong et al. [21] wrote fiber grating with laser and the experiments have confirmed that the fiber grating written by this method exhibits higher thermal stability. The refractive index modulation written by UV laser is very small (10^−4^ or 10^−3^ magnitude), and it is easily erased when the operating temperature is higher than 400 °C [22,23,24]. This phenomenon is called the thermal degradation effect of the fiber grating.

In order to measure higher temperatures, the annealing process [26,27] is used to make regenerative FBG. Jun He et al. [26] prepared an FBG with negative index and a temperature range of up to 1000 °C, based on thermal regeneration. Tyson L et al. [27] have developed a high temperature resistant fiber grating, which can withstand high temperature up to 1100 °C and has good linearity. High temperature annealing brings the feature of high temperature resistance, but the problem of insufficient strength still needs to be overcome. To solve this problem, high temperature resistant femtosecond pure silica FBG gradually becomes a research focus. Venkata Reddy Mamidi et al. [28] configured a FBG by encapsulating a femtosecond laser and measured the temperature from 20 °C to 650 °C with a resolution of 1 °C. Ding Xudong et al. [29] from Beijing information science and technology university have studied the temperature characteristic of pure quartz FBG, which present good linearity of 0.9996 between 20 °C and 1000 °C. Based on the above content, compare and organize the characteristics of various FBGs, as shown in Table 1. It is found that the femtosecond pure silica FBG has the advantages of stable structure and high temperature resistance under extreme environments, so it is suitable to be used for high temperature sensing.

In the process of acupuncture, the fluid of the human body can be contacted [30,31]. The main electrolytes in body fluids are Na^+^, Cl^−^, K^+^, SO4^−^, Ca^2+^, ${HPO}_{4}^{2-}$, Mg^2+^ and ${HCO}_{3}^{-}$, as well as some organic acids and proteins. Their content affects the refractive index of body fluids. By adding optical fiber sensor to the hollow fire needle, the body temperature in the needle can be monitored more accurately, and at the same time, it can be determined whether the balance of organic acids and proteins in the body fluid is normal. Judge by monitoring the refractive index of body fluid. In addition, the optical fiber refractive index sensor has the ability to expand. Through functional surface modification [32], specific proteins (antigens) [33], hydrophilic gels [34], etc., the measurement of refractive index can be converted into virus [35], DNA [36], humidity [37], etc., it is necessary to use optical fiber sensors to measure refractive index.

As for dual-parameter measurement, two temperature refractive index probes are often required to detect separately, which will result in low accuracy of the position of the probed point. Therefore, how to implement a sensor head to measure the temperature and refractive index of a point at the same time has certain significance. Most temperature responses have obvious wavelength drift characteristics. If the refractive index is judged only by the wavelength change, the dual parameters cannot be demodulated. Therefore, in order to achieve multiparameter measurement, we introduce intensity measurement. The characteristics of light waves include intensity and wavelength, by monitoring the wavelength and intensity dual independent variables, the temperature and refractive index can be measured simultaneously on an FBG structure, and refractive index researcher pay more attention to intensity type. For example, Kyung Rak Sohn et al. [38] developed an all-fiber-optic sensors, single-mode and multimode fibers were used to fabricate sensing probe, and the experiment obtained a sensitivity of 50 dB/RIU in the range of 1.33–1.47. Cheng Zhang et al. [39] proposed an intensity fiber optic sensor based on double FBGs cascade structure; the refractive index response sensitivity is segmented; in the range of 1.3326–1.3702 the refractive index response sensitivity is –199.6 dB/RIU; and in the range of 1.3702–1.4066, the refractive index response sensitivity is –355.5 dB/RIU. Wen Zhang et al. [40] proposed a low reflectivity interferometer based on FBG and Fabry–Perot interferometer; the temperature sensitivities based on FBG peak and PCF-FP dip in the range of 30–120 °C are 11.46 pm/°C and 8.62 pm/°C, respectively. The refractive index sensitivities based on FBG peak and PCF-FP dip in the range of 1.3315–1.3708 are 0 and 9.14 dB/RIU, respectively. In 2019, Yue Wu et al. [41] developed a Fabry–Perot interferometer based on FBG; the fiber sensor has the temperature sensitivity of 9 pm/°C in the range from 10 °C to 70 °C, and refractive index sensitivity is 33.55 dB/RIU in the range from 1.333 to 1.413. However, although these sensors have completed dual-parameter measurement, they still need to optimize the problems of complex structure and small range. It is necessary to further study dual-parameter optical fiber sensor with small structure and large measuring range.

In this paper, a dual-parameter sensor used to measure temperature and refractive index was produced. By comparing the material characteristics of different sensor designs before and after high temperature, a high temperature resistant FBG was obtained by laser writing, and the sensor’s refractive index sensitivity was improved by chemical etching. The sensor’s temperature and refractive index measurement was theoretically analyzed, verified by experiments. As the temperature and refractive index change, the spectral peak wavelength and intensity will correspondingly change. The experimental results show that the smart needles made in this way can combine classical acupuncture with modern technology, thereby optimizing the acupuncture needle burning process and the measurement process. It is more conducive to the improvement of medical standards and contributes to the integration of medicine and engineering.

## 2. Materials and Methods

### 2.1. Material Selection and Analysis

The material properties of glass optical fiber FBG, regenerated FBG and pure silica FBG before and after high temperature are analyzed. The three kinds of fiber grating materials at room temperature and 1000 °C were put into X-ray diffraction (XRD) (Xpert Pro, Rotterdam, The Netherlands), and various materials were analyzed in detail using monochromatic CuKα (λ = 1.54 Å). The angle between the incident angle and the reflection angle of X-rays is 2θ.

Compare material properties at room temperature and after a high temperature of 1000 °C, the glass FBG material will transform from amorphous to cristobalite crystals. The peak strength is different, and the material composition changes. As shown in Figure 1a, the calcite is precipitated. The strongest diffraction peak occurs at 29.1° matching the first peak of cristobalite SiO_2_ crystal face (111) (PDF No. 82–1406) and 31.7° matching the second peak, which are consistent with the crystal faces of cristobalite (102). The result shows that glass optical fiber produces degradation phenomenon, which makes the spectrum disappear, and this material is not suitable for high temperature measurement.

After thermal annealing at high temperature, the crystal image of the regenerative FBG will change significantly, the XRD pattern is shown in Figure 1b. It can be noted that the crystal structure of optical fiber after annealing treatments significantly different from the optical fiber which is amorphous. For the fiber after 1000 °C annealing steps, its crystal structure is SiO_2_, and the strongest diffraction peak occurs at 21.2° matching the first peak of SiO_2_ crystal face (100) (PDF No. 33–1161). Some weak peaks are also observed, which are consistent with the crystal faces of SiO_2_ (101), (110), (112), (210), respectively. It indicates that new crystal (SiO_2_) has been formed after annealing. Although the manufactured sensor has a stable composition distribution, the sensor material has changed from before, and the mechanical strength has decreased, which is not conducive to use as a high-temperature material.

Pure silica FBG is a sensor based on silica glass as the core material. The core and cladding materials of pure silica fiber are both silica glass, but the refractive index of the core silica glass is higher than that of the cladding silica glass through doping technology. Its chemical composition is mainly doped with Na_2_SiO_3_, CaSiO_3_, Na_2_O·CaO·6SiO_2_ and other components. In the process of heating to 1000 °C and cooling to room temperature, the XRD diagram of the material composition can be shown in Figure 1c. With the temperature rising, the material composition of the material is relatively fixed. This is the main reason why the pure quartz FBG can work steadily for a long time at high temperature. Compared with other materials, pure quartz FBG can maintain a more stable amorphous state after high temperature, and the material composition has no changes. The material has high stability, does not crystallize like glass optical fibers, and can still ensure a certain degree of light transmittance at high temperatures. Moreover, pure silica fiber grating is different from regenerative FBG, and it still has a certain mechanical strength after high temperature. Therefore, pure silica fiber is a good optical fiber material that can withstand high temperatures and can be used to make fiber optic sensor in high temperature environment.

### 2.2. Measurement Method of Temperature Wavelength Sensor

The transmission process of light in the FBG is shown in Figure 2. The optical transmission mode is mainly represented by transmitted light propagating forward and reflected light propagating backward in the core.

The Bragg equation of fiber grating is:(1)$$\lambda_{B}=2n_{eff}\Lambda$$

When the temperature changes, due to thermo-optical and thermal expansion effects, the effective refractive index *n_eff_* and grating period Λ will change accordingly. This will cause the center wavelength $\lambda_{B}$ change. Performing differential decomposition on Equation (1), we can get Equation (2):
(2)$$\begin{array}{l} \Delta \lambda_{B}=2(\Lambda \frac{\partial n_{\mathrm{eff}}}{\partial a}+n_{\mathrm{eff}}\frac{\partial \Lambda }{\partial a})\Delta a+2(\Lambda \frac{\partial n_{\mathrm{eff}}}{\partial T}+n_{\mathrm{eff}}\frac{\partial \Lambda }{\partial T})\Delta T=2\left(\Lambda \frac{\partial n_{\mathrm{eff}}}{\partial a}\Delta a+n_{\mathrm{eff}}\frac{\partial \Lambda }{\partial a}\Delta a+\Lambda \frac{\partial n_{\mathrm{eff}}}{\partial T}\Delta T+n_{\mathrm{eff}}\frac{\partial \Lambda }{\partial T}\Delta T\right)\\ =2\left(\Lambda \frac{\partial n_{\mathrm{eff}}}{\partial T}\Delta T+\Lambda \frac{\partial n_{\mathrm{eff}}}{\partial a}\Delta a+n_{\mathrm{eff}}\frac{\partial \Lambda }{\partial a}\Delta a+n_{\mathrm{eff}}\frac{\partial \Lambda }{\partial T}\Delta T\right)=2\left(\left[\frac{\partial n_{\mathrm{eff}}}{\partial T}\Delta T+\frac{\partial n_{\mathrm{eff}}}{\partial a}\Delta a+\frac{n_{\mathrm{eff}}}{\Lambda }\frac{\partial \Lambda }{\partial a}\Delta a\right]\Lambda +n_{\mathrm{eff}}\frac{\partial \Lambda }{\partial T}\Delta T\right)\\ =2\left\{\left[\frac{\partial n_{\mathrm{eff}}}{\partial T}\Delta T+{\left(\Delta n_{\mathrm{eff}}\right)}_{ep}+\frac{\partial n_{\mathrm{eff}}}{\partial a}\Delta a\right]\Lambda +n_{\mathrm{eff}}\frac{\partial \Lambda }{\partial T}\Delta T\right\} \end{array}$$

In the equation, (Δ*n*_*eff*_)_*ep*_ represents the elastic-optical effect caused by thermal expansion, $\frac{\partial \Lambda }{\partial T}$ is the thermal expansion coefficient of Λ and $\frac{\partial n_{\mathrm{eff}}}{\partial a}$ is the waveguide effect caused by the thermal expansion. In the temperature experiment, when the center wavelength of the sensor is known, the drift of the wavelength relative to the center wavelength is as shown in Equation (3), which can be obtained by Equation (1) and Equation (2). When the material is constant, the changes in refractive index, waveguide structure and grating period due to thermo-optical effects and thermal expansion effects are constant. Therefore, the wavelength shift is positively related to temperature.
(3)$$\begin{array}{l} \frac{\Delta \lambda_{B}}{\lambda_{B}}=\frac{1}{n_{\mathrm{eff}}}\frac{\partial n_{\mathrm{eff}}}{\partial T}\Delta T+\frac{1}{n_{\mathrm{eff}}}\frac{{\left(\Delta n_{\mathrm{eff}}\right)}_{\mathrm{ep}}}{\Delta T}\Delta T+\frac{1}{n_{\mathrm{eff}}}\frac{\partial n_{\mathrm{eff}}}{\partial a}\frac{\Delta a}{\Delta T}\Delta T+\frac{1}{\Lambda }\frac{\partial \Lambda }{\partial T}\Delta T\\ \left[\frac{1}{n_{\mathrm{eff}}}\frac{\partial n_{\mathrm{eff}}}{\partial T}+\frac{1}{n_{\mathrm{eff}}}\frac{\partial {\left(\Delta n_{\mathrm{eff}}\right)}_{\mathrm{ep}}}{\partial T}+\frac{1}{n_{\mathrm{eff}}}\frac{\partial n_{\mathrm{eff}}}{\partial a}\frac{\Delta a}{\Delta T}+\frac{1}{\Lambda }\frac{\partial \Lambda }{\partial T}\right]\Delta T \end{array}$$

### 2.3. Measuring Method of Refractive Index Intensity Type Sensor

The intensity change of the sensor is based on the reflection coefficient at the boundary between two different media. Figure 3 shows that the wave incident on the plane boundary is partially transmitted and reflected. As shown in Figure 4, The process of Finier reflection on the grating is simulated by BPM software. Among them, the yellow part is the fiber cladding, the red part is the fiber core part, the light green part is the FBG, and the dark green part is the environment to be tested. The incident light enters the grating from bottom to top, and a strong Fresnel reflection occurs on the end face of the fiber optic sensor.

Observed from the side, it is found that the intensity distribution on the grating structure has high bright spots, which characterizes the strong reflection of the grating part. In addition, Figure 5b,c are the intensity distribution of the transmitted light before and after the grating, respectively. It can be seen from the light intensity distribution on the end surface that after being reflected by the grating, a small amount of transmitted light will pass through, and the end surface will be reflected again and transmitted to the external environment to be lost.

Analyze the light passing through the grating, as shown in Figure 5. Due to the mismatch of refractive index, the reflected light intensity of the sensor will change with the change of the external environment, and then an intensity type refractive index sensor is obtained. In order to improve the sensitivity of the sensor, we add the refractive index loss on the side of the sensor, as shown in Figure 6.

Further analysis of the sensor probe shows that when the light propagates in a short distance, it will produce higher-order modes and mainly propagate in the cladding. As shown in Figure 7, based on the optical analysis software, combined with our structural design, the fiber probes with core and cladding diameters of 10 μm and 60 μm, and refractive indexes of 1.4681 and 1.4544 were analyzed. Based on the light refraction law, the refractive index of the external environment will affect the loss of high-order modes, and then realize the intensity response of the sensor.

## 3. Production and Packaging

In the experimental design, pure quartz fiber is selected as the basic material, and femtosecond writing method is used to write FBG with stable sensor structure. Secondly, hydrofluoric acid (HF) is used to corrode the fiber probe to improve the sensitivity of refractive index response of the sensor. Thirdly, a cutter is used to cut one end of the sensor to increase the sensitivity of refractive index response of the sensor. The end face of the probe after cutting is detected and analyzed to ensure the flatness of the end face. Finally, the sensor is packaged in the cavity of the hollow needle, and cured by adding high temperature glue. The production and fixation of the sensor are achieved through the above steps, as shown in Figure 8.

### 3.1. Laser Writing

Figure 9 is the laboratory photo, because the geometry of the fiber itself is cylindrical, the bent cladding and core structure will defocus the focal point of the femtosecond laser during the femtosecond direct writing process, which greatly reduces the power of the focus area. It can even cause problems such as the FBG cannot be written. In this process, a refractive index matching liquid with a higher numerical aperture can improve the defocusing problem of the light spot. When the laser is focused on the fiber core through the objective lens, because the refractive index of the refractive index matching liquid is similar to that of the fiber cladding, the effect of the cylindrical lens on the fiber surface can be better reduced.

The specific steps are as follows: Firstly, a low-power objective lens is used to find the fiber cladding, and a high-power objective lens with high numerical aperture is used to find the fiber core. When the fiber core is observed clearly under charge coupled device (CCD), the actual focus of femtosecond laser is usually near 1 μm below the plane because the CCD focus and femtosecond system focus are not consistent. By adjusting the position of the focusing plane, the written refractive index modulation point is located in the central region of the fiber core. After the femtosecond laser beam is focused by the objective lens, the beam waist radius is about 0.5 μm, and the written spot diameter is about 1 μm. According to Equation (3), we chose to write the second order point by point FBG with a period of 1.112 μm and a length of about 4 mm.

### 3.2. Etching of the Sensor and End Face Cutting

It is also necessary to chemically etch the fiber cladding to improve the refractive index sensitivity of the sensor. As shown in Formulas (4) and (5), when the concentration is in the range of 2% to 24%, two etching reactions take place simultaneously. The SiF_4_ produced is not a stable substance, which will produce fluorosilicic acid when it meets with water. Because only the second reaction takes place at high concentration, 40% HF is selected in this experiment.
(4)$$SiO_{2}+4HF\to SiF_{4}+2H_{2}O$$
(5)$$SiO_{2}+6HF\to H_{2}SiF_{4}+2H_{2}O$$

Considering the fluidity of HF solution, glycerol is dripped on both ends of the fiber in order to prevent the acid from diffusing along the fiber to the non-corrosive region and corroding other fibers. Because glycerol is viscous, small mobility, and because glycerol and HF reaction is slow, it can effectively prevent HF diffusion to the coupling point and input and output fiber, thus avoiding etching to the part outside the sensor.

Then, at a room temperature of 20 °C, use a pipette to draw 40% HF solution onto the glass slide and record the dripping time. After 20 min, use a needle to quickly suck away the HF and quickly drop a 10% NaOH solution (high concentration will react violently to break the thin optical fiber) to neutralize the remaining HF. Suck away the solution in the tube and wash the optical fiber with clean water. The photo of the optical fiber taken before and after etching is shown in Figure 10. The diameter of optical fiber changed from 125 μm to 60 μm by etching.

### 3.3. Fixing of Sensor Probe

In the sensor probe, there are two parts that directly contact the external environment: the temperature sensor and package structure part. The sensor is located at the front end of the sensor probe, used to sense the temperature and refractive index. In order to enable the sensor probe to contact the external environment, the package structure need to be designed as a through-hole structure. In this paper, a hollow fire needle is used as the package structure.

Fire needle is a needle with a name, from “Preparation of Emergency Medicine”. Now it is mostly made of stainless steel, and the hot needle is usually a hollow needle with a diameter of 0.4 to 0.6 mm. The appearance of the hollow needle is basically the same as that of ordinary acupuncture needles, and 5 to 1.5 inches are more commonly used. The needle body is thick and round, sharp, heat dissipating quickly, and difficult to deform. When using, burn the needle to red, and puncture the selected part quickly. Coat the surface of the optical fiber with liquid high temperature glue, put the sensor in the cavity of the hollow needle, and use the glue to fix the sensor through high temperature curing. Based on the control of the etching time, the size of the sensor is changed to 60 μm. By adjusting the height of the cutter, the thin optical fiber can be cut smoothly, and the cutting effect can be checked by the End-face Detection. As shown in Figure 11, first apply glue on a section of the lead-out optical fiber at the back of the sensor probe; then put the optical fiber into the hollow needle, and fill the end of the hollow needle with glue after fixing the optical fiber, and finally wait for the sensor and the hollow needle to be tightened before measuring. Because the sensing probe is not fixed with glue, it is structurally fixed by the conductive fiber and the inner wall of the hollow needle. Therefore, the stress mainly acts on the lead-out fiber, and the sensor has no obvious influence.

## 4. Results

### 4.1. Temperature Measurement

To calibrate the temperature characteristics of the sensor, a temperature measurement test platform is established as shown in Figure 12, which is composed of a broad band light source (spectral range: 1500–1600 nm, BBS) and an optical spectrum analyzer (OSA) (wavelength resolution: 0.02 nm). The light output from BBS is transmitted to the sensor, and then some reflected light is reflected back from the transmission fiber to OSA.

The sensor is placed in a furnace to measure temperature characteristics. As the furnace temperature changes from 30 °C to 1150 °C, the spectra of the FBG will drift with different temperature. Figure 13 is the high temperature experiment site in laboratory.

The sensor is placed in a furnace to study the thermal effect. Figure 14 is the wavelength shift with the increase of temperature. The measured data are represented by the dots, and the fitting line are obtained by using least squares linear fit, as shown in Figure 15. The experimental results show that the temperature response sensitivity is 0.0134 nm/°C with a good linearity, and the influence of temperature on the spectral intensity is not obvious. As a result, the external temperature can be detected based on the wavelength peak drift of the spectrum.

Subsequently, the dynamic characteristics of the sensor are measured. A temperature calibration experiment was carried out at the Beijing Great Wall Metrology and Testing Technology Research Institute of Aviation Industry Corporation of China (Beijing, China). The name of the equipment used for calibration is the RD-05 thermal calibration wind tunnel (304 Research Institute, Beijing, China), which has a temperature measurement range up to 1100 °C, and a Mach number range of 0.2 Mach to 0.9 Mach. The working temperature values are around 30.1 °C, 890.2 °C, 1100.8 °C, respectively.

Finally, the institute issued a calibration report of the sensor at high temperature to characterize that the sensor can withstand a high temperature of 1100 °C. Table 2 lists the calibration results of high temperature measurement deviation.

Based on the above experiment, we use the manufactured sensor for the needle burning process. As shown in Figure 16, the experiment will conduct experimental research on the different states of acupuncture needles during needle burning, such as normal temperature, inner flame, middle flame and outer flame. The experimental results are as shown in Figure 17.

The experimental results are shown in Table 3. By monitoring the wavelength of the reflection peak and referring to the sensitivity of the temperature response, the temperature value in different environments can be obtained.

### 4.2. Refractive Index Measurement

As mentioned in the introduction, refractive index measurement makes a great contribution to medical research. In addition, if the change in refractive index is wavelength type, it will affect the accuracy of temperature measurement. Therefore, the study of the refractive index characteristics of the sensor is also very necessary. In the experiment, different concentrations of sucrose solution are configured as refractive index samples, and the refractive indexes are 1.351, 1.357, 1.369, 1.3788, 1.3872, 1.3941, 1.3872 after being tested by Abbe refractive index detector (Mince Instrument, Fujian, China). The experimental device diagram is shown in Figure 18. The light emitted by the BBS is incident on the sensor, and the MS740A spectrometer (Anritsu, Tokyo, Japan) is used to measure the reflectance spectrum. During the refractive index measurement process, put liquids with different refractive indices into the container, and the sensor is immersed in the liquid. After completing a measurement, clean the sensor with absolute ethanol after each measurement, and dry it until the response spectrum is consistent with the original spectrum of the sensor in the air, then proceed to the next refractive index experiment.

The spectra of sensor in different refractive index solutions are measured experimentally. When the refractive index of analytes changes, the intensity of sensor spectrum will change. The refractive index can be measured by measuring the peak intensity. In the experiment, the peak at the wavelength of 1554.15 nm was selected as the monitoring object, and its intensity changes under different refractive indexes were recorded. It can be seen from the Figure 16 that when the refractive index of the ambient liquid varies from 1.333 to 1.4027, the power of the reflection peak decreases with the increase of the external refractive index.

The real points in Figure 19 are experimental data, and the solid lines are linear fitting lines. With the increase of external refractive index, the spectral peak intensity decreases and the refractive index response sensitivity is −49.044 dB/RIU. In addition, the wavelength response to the refractive index is studied, and the experiment found that when the refractive index changes greatly, the center wavelength of the interference spectrum will also move to a certain extent, and the response sensitivity is 0.053 nm/RIU. We also studied the refractive index response of the sensor, and obtained the change of spectral intensity and wavelength with refractive index, as shown in Figure 20.

In order to further verify the reliability of the hollow acupuncture measurement system in practical applications, the fish blood was taken out and put into a test tube. The blood concentration is changed by adding inorganic salt and purified water, and the concentration of the blood sample is calibrated with a refractometer, as shown in Figure 21. Insert the hollow needle with the optical fiber sensor into the blood of different concentration to realize the measurement of the refractive index of the fish blood reagent under the different concentration.

The experimental results are shown in Figure 22. Under the condition of a certain temperature, the sensor performs intensity response test on blood samples with different refractive indexes. By monitoring the intensity of the reflection peak and referring to the sensitivity of the refractive index response, the refractive index value of the blood sample can be obtained, as shown in Table 4. Through the biological blood, certain data support can be provided for medical diagnosis and treatment.

## 5. Discussion

### 5.1. Temperature Response Characteristics of Sensor Wavelength

The wavelength change is related to the wavelength band, the thermo-optical coefficient and the thermal expansion coefficient. When the temperature changes, the grating period and effective refractive index will change, which will cause the wavelength shift. Equation (3) can be written as
(6)$$\Delta \lambda_{B}=\left[\xi +\alpha +\frac{1}{n_{\mathrm{eff}}}\frac{\partial {\left(\Delta n_{\mathrm{eff}}\right)}_{\mathrm{ep}}}{\partial T}+\frac{\partial n_{\mathrm{eff}}}{\partial a}\frac{\Delta a}{\Delta T}\right]\lambda_{B}\Delta T=\eta \lambda_{B}\Delta T$$
among them, ξ represents the thermo-optical coefficient of fiber grating $\frac{1}{n_{\mathrm{eff}}}\frac{\partial n_{\mathrm{eff}}}{\partial T}$. Means, α represents the thermal expansion coefficient $\frac{1}{\Lambda }\frac{\partial \Lambda }{\partial T}$.η is the temperature sensitivity coefficient of FBG and can be written as
(7)$$\eta =\xi +\alpha +\frac{1}{n_{\mathrm{eff}}}\frac{\partial {\left(\Delta n_{\mathrm{eff}}\right)}_{\mathrm{ep}}}{\partial T}+\frac{1}{n_{\mathrm{eff}}}\frac{\partial n_{\mathrm{eff}}}{\partial a}\frac{\Delta a}{\Delta T}$$
when the material is determined, η is a positive constant related to the material coefficient basically. The center wavelength of the sensor in the experiment is also a positive number at 1554.15 nm. Based on Formula (6), it can be known that the center wavelength shifts to the long wave direction as the temperature increases. The theoretical analysis is consistent with the experimental results. Because of the effect of elasto-optical effect is smaller than the thermo-optical coefficient, so they can completely ignore their impact on temperature sensitivity coefficient. Therefore, Equation (7) can be written as Equation (8), which can be used to explain that the sensor has a good wavelength linear output in the experiment. Theory and experiment can correspond reasonably.
(8)η=ξ+α

### 5.2. Refractive Index Response Characteristics of Sensor Intensity

When the refractive index changes, the response of the sensor’s intensity is mainly composed of the side high-order light loss and the end-face transmitted light loss. In the proposed sensor, the reflection coefficients at the boundary between two different dielectric media are different. When the cladding of the optical fiber is corroded, due to the change of the waveguide structure, some light of the cladding restricted by the cladding enters the surrounding medium. Before explaining the principle of the proposed sensor, the reflection coefficients for waves that are polarized perpendicular (s-polarized) and parallel (p-polarized) to the plane of incidence can be defined in advance. For side light loss, the longitudinal loss of higher-order modes is represented by r_s_. The loss of higher-order modes in the lateral direction is represented by r_p_. With the change of the external refractive index, some light in the optical fiber does not meet the reflection conditions in the solution [38,42]. The reflectance equation can be written as:(9)$$r_{s}=\frac{-n_{2}^{2}\cos\theta_{1}+n_{1}\sqrt{\left(n_{2}^{2}-n_{1}^{2}\sin^{2}\theta_{1}\right)}}{n_{2}^{2}\cos\theta_{1}+n_{1}\sqrt{\left(n_{2}^{2}-n_{1}^{2}\sin^{2}\theta_{1}\right)}}$$
(10)$$r_{p}=\frac{n_{1}\cos\theta_{1}-\sqrt{\left(n_{2}^{2}-n_{1}^{2}\sin^{2}\theta_{1}\right)}}{n_{1}\cos\theta_{1}+\sqrt{\left(n_{2}^{2}-n_{1}^{2}\sin^{2}\theta_{1}\right)}}$$

Coupled with the cutting of one end of the sensor, the end face of the optical fiber will also produce end-face light loss, as shown in the following formula.
(11)$$r_{s}{}^{\prime }=\frac{-n_{2}^{2}\cos\theta_{1}+n_{0}\sqrt{\left(n_{2}^{2}-n_{0}^{2}\sin^{2}\theta_{2}\right)}}{n_{2}^{2}\cos\theta_{1}+n_{0}\sqrt{\left(n_{2}^{2}-n_{0}^{2}\sin^{2}\theta_{2}\right)}}$$
(12)$$r_{p}{}^{\prime }=\frac{n_{1}\cos\theta_{2}-\sqrt{\left(n_{2}^{2}-n_{0}^{2}\sin^{2}\theta_{2}\right)}}{n_{1}\cos\theta_{2}+\sqrt{\left(n_{2}^{2}-n_{0}^{2}\sin^{2}\theta_{2}\right)}}$$
where *n*_0_, *n*_1_ and *n*_2_ represent the refractive indices of optical fiber core, optical fiber clading and external refractive index, respectively. Where *n*_0_ = 1.4681, *n*_1_ = 1.4544. The intensity of the reflected light I_r_ depends on the external refractive index *n*_2_. It can be expressed as the following formula:(13)$$\begin{array}{l} I_{r}=I_{0}-I_{0}\times r=I_{0}-I_{0}\times \left(r_{s}+r_{p}+r_{s}{}^{\prime }+r_{p}{}^{\prime }\right)\\ =I_{0}-I_{0}\times \left[\begin{array}{c} \frac{-n_{2}^{2}\cos\theta_{1}+n_{1}\sqrt{\left(n_{2}^{2}-n_{1}^{2}\sin^{2}\theta_{1}\right)}}{n_{2}^{2}\cos\theta_{1}+n_{1}\sqrt{\left(n_{2}^{2}-n_{1}^{2}\sin^{2}\theta_{1}\right)}}+\frac{n_{1}\cos\theta_{1}-\sqrt{\left(n_{2}^{2}-n_{1}^{2}\sin^{2}\theta_{1}\right)}}{n_{1}\cos\theta_{1}+\sqrt{\left(n_{2}^{2}-n_{1}^{2}\sin^{2}\theta_{1}\right)}}\\ +\frac{-n_{2}^{2}\cos\theta_{2}+n_{0}\sqrt{\left(n_{2}^{2}-n_{0}^{2}\sin^{2}\theta_{2}\right)}}{n_{2}^{2}\cos\theta_{2}+n_{0}\sqrt{\left(n_{2}^{2}-n_{0}^{2}\sin^{2}\theta_{2}\right)}}+\frac{n_{1}\cos\theta_{2}-\sqrt{\left(n_{2}^{2}-n_{0}^{2}\sin^{2}\theta_{2}\right)}}{n_{1}\cos\theta_{2}+\sqrt{\left(n_{2}^{2}-n_{0}^{2}\sin^{2}\theta_{2}\right)}} \end{array}\right]\\ \approx I_{0}\times \left[1-\frac{-n_{2}^{2}\cos\theta_{1}+n_{1}\sqrt{\left(n_{2}^{2}-n_{1}^{2}\sin^{2}\theta_{1}\right)}}{n_{2}^{2}\cos\theta_{1}+n_{1}\sqrt{\left(n_{2}^{2}-n_{1}^{2}\sin^{2}\theta_{1}\right)}}-\frac{n_{1}\cos\theta_{1}-\sqrt{\left(n_{2}^{2}-n_{1}^{2}\sin^{2}\theta_{1}\right)}}{n_{1}\cos\theta_{1}+\sqrt{\left(n_{2}^{2}-n_{1}^{2}\sin^{2}\theta_{1}\right)}}-\frac{-n_{2}^{2}+n_{0}n_{2}^{}}{n_{2}^{2}+n_{0}n_{2}^{}}-\frac{n_{1}-n_{2}^{}}{n_{1}+n_{2}^{}}\right] \end{array}$$


I_0_ represents the incident light intensity, which can be set to unit 1. *θ*_2_ is the incident angle of light parallel to the optical axis. Based on the limitation of the numerical aperture, the incident angle is very small and can be approximated to 0. *θ*_1_ is the light incident angle perpendicular to the optical axis. Because the lost light intensity is mainly concentrated in a small angle range, we have analyzed small angles ranging from 0 to 50°. Through MATLAB software simulation, we can get the change of intensity with external refractive index at different angles based on Formula (13), as shown in Figure 23.

When the three-dimensional simulation diagram of Figure 23 is turned to the intensity as the ordinate and the external refractive index as the abscissa. The intensity of the reflected light decreases with the increase of the external refractive index, and the spectral intensity changes linearly with the refractive index, as shown in Figure 24. The theoretical simulation is consistent with the experimental results.

### 5.3. Two-Parameter Simultaneous Measurement Analysis

When the temperature and refractive index of the environment to be measured change at the same time. We can sort out the temperature and refractive index response of the sensor, and get the spectral characteristic signal changes with temperature and refractive index [43].
(14)$$\left[\begin{array}{c} \Delta \lambda \\ \Delta I \end{array}\right]=\left[\begin{array}{cc} K_{T\lambda } & K_{n\lambda }\\ K_{TI} & K_{nI} \end{array}\right]\left[\begin{array}{c} \Delta T\\ \Delta n \end{array}\right]$$

Equation (14) can be written as:(15)$$\left[\begin{array}{c} \Delta T\\ \Delta n \end{array}\right]={\left[\begin{array}{cc} K_{T\lambda } & K_{n\lambda }\\ K_{TI} & K_{nI} \end{array}\right]}^{-1}\left[\begin{array}{c} \Delta \lambda \\ \Delta I \end{array}\right]$$

With the response sensitivity data, the temperature and refractive index of the measurement point can be obtained from sensitivity matrix equation, as shown in Equation (16).
(16)$$\left[\begin{array}{c} \Delta T\\ \Delta n \end{array}\right]={\left[\begin{array}{cc} 0.0134 & -0.00001\\ 0.053 & -49.044 \end{array}\right]}^{-1}\left[\begin{array}{c} \Delta \lambda \\ \Delta I \end{array}\right]$$

Based on the above analysis, we performed acupuncture on fish to complete the verification of Two-parameter simultaneous measurement. As shown in Figure 25, the experiment used hollow acupuncture needles to measure the belly and gill of fish. Because the fish is taken out from the water, we have also conducted experimental research on water. The experimental results are shown in Figure 26, the refractive index of the gill is slightly larger than the abdomen. Therefore, when the acupuncture needle is pierced into the fish, the reflection wavelength will move based on the temperature and refractive index accordingly.

By monitoring the reflectance spectrum and referring to the sensitivity equation, the refractive index value and temperature can be obtained, as shown in Table 5. In the experiment, the water temperature was 8 °C, the fish maw temperature was 10 °C, and the fish gill temperature was 12 °C. In addition, the refractive index of fish gills is the largest at 1.387, the refractive index of fish maw is 1.379, and the refractive index of water is 1.333. The position of the needle can be roughly distinguished by the refractive index.

## 6. Conclusions

A compact optical fiber sensing structure for temperature and refractive index measurement is proposed in this paper. Femtosecond laser writing method is selected to write pure silica fiber to improve temperature tolerance. At the same time, the FBG is corroded to realize the measurement of the refractive index. In the temperature range up to 1150 °C, the temperature sensitivity can be maintained at 0.0134 nm/°C, and the refractive index sensitivity from 1.333 to 1.4027 is −49.044 dB/RIU. Based on sensitivity matrix equation, the dual-parameter detection of temperature and refractive index is realized. In addition to the improvement of the functions of the acupuncture needle, this kind of temperature-refractive index dual-parameter measurement has great application prospects in the field of biomedicine, and it is one of the breakthroughs in the integration of medicine and engineering.

## Figures and Tables

**Figure 1 materials-14-01028-f001:**
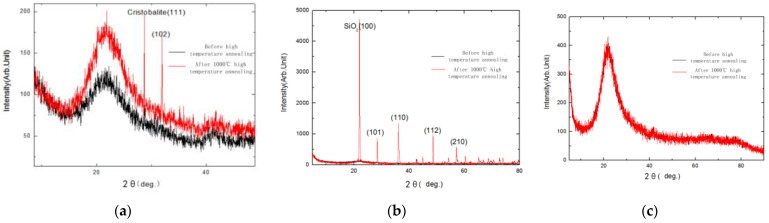
X-ray analysis before and after high temperature: (**a**) glass FBG; (**b**) regenerative FBG; (**c**) pure quartz FBG.

**Figure 2 materials-14-01028-f002:**
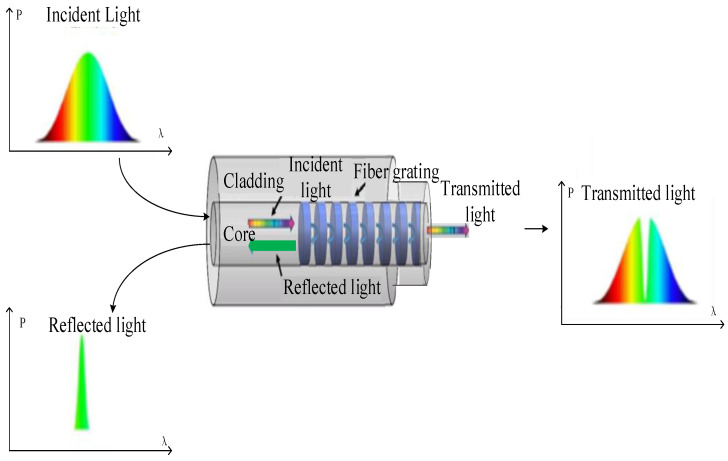
Spectral characteristics of fiber grating sensing.

**Figure 3 materials-14-01028-f003:**
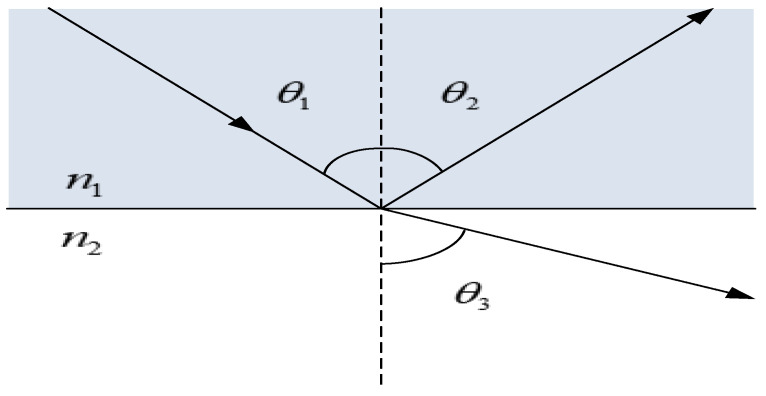
Reflection and refraction of light at the interface.

**Figure 4 materials-14-01028-f004:**
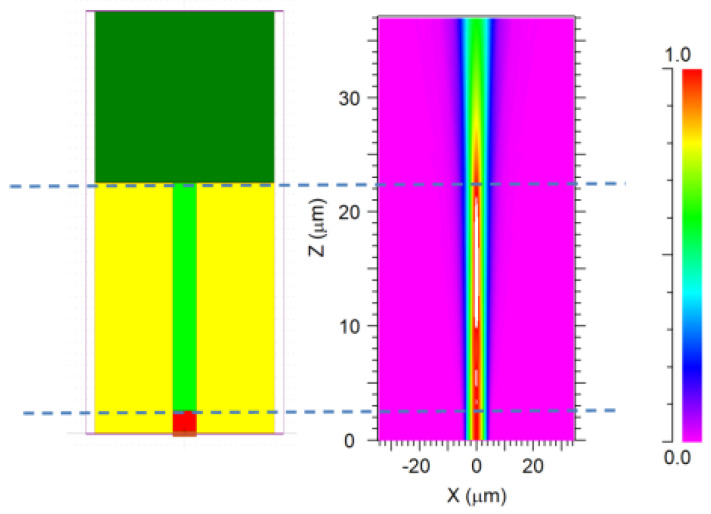
Fresnel reflection analysis based on BPM software.

**Figure 5 materials-14-01028-f005:**
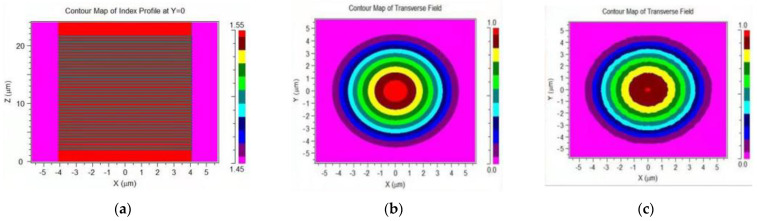
Simulate the light intensity change on the fiber cross section based on BPM software: (**a**) grating structure design; (**b**) light intensity distribution at the entrance of the grating; (**c**) light intensity distribution at the exit of the grating.

**Figure 6 materials-14-01028-f006:**
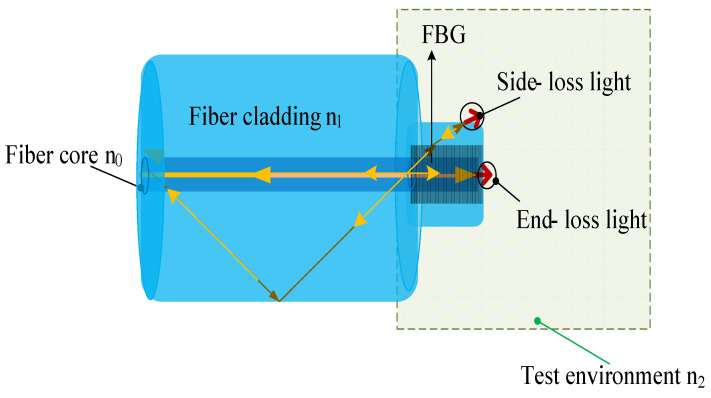
The loss analysis of optical fiber sensing probe.

**Figure 7 materials-14-01028-f007:**
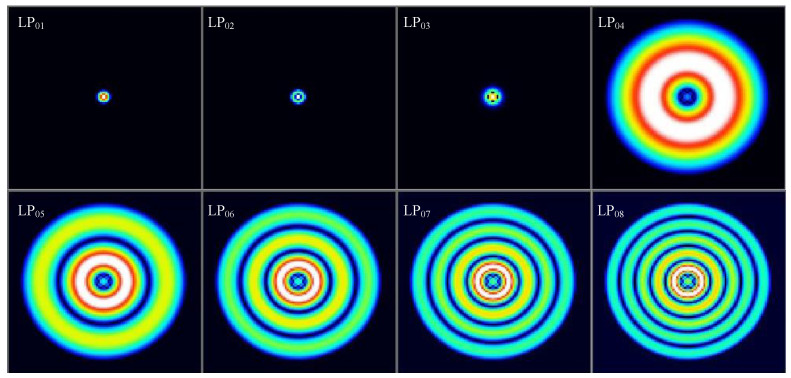
Mode field distribution of different modes.

**Figure 8 materials-14-01028-f008:**
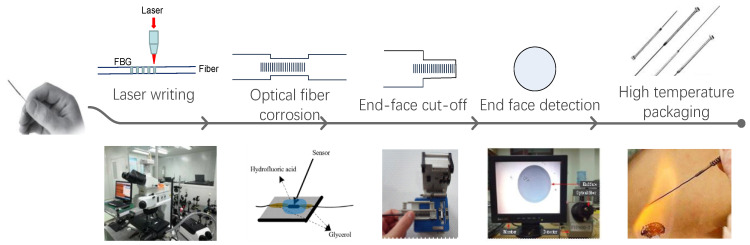
Production flow chart of optical fiber sensor.

**Figure 9 materials-14-01028-f009:**
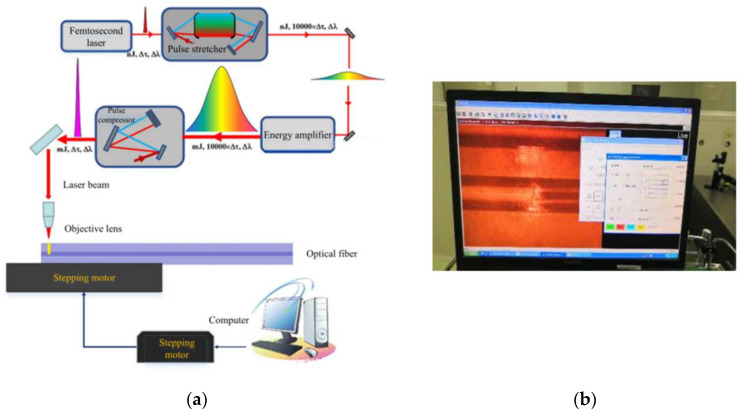
Inscribe fiber grating with femtosecond: (**a**) Schematic diagram of grating writing structure; (**b**) Monitoring interface during the writing process.

**Figure 10 materials-14-01028-f010:**
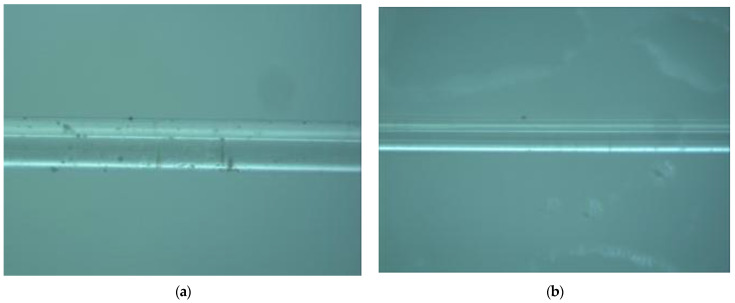
Optical fiber sensor before and after the etching: (**a**) Optical fiber before etching 125 μm; (**b**) Optical fiber after etching 60 μm.

**Figure 11 materials-14-01028-f011:**
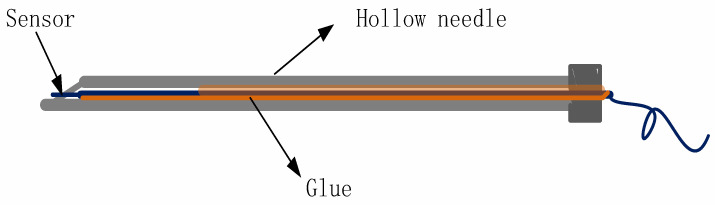
Schematic diagram of sensor package structure.

**Figure 12 materials-14-01028-f012:**
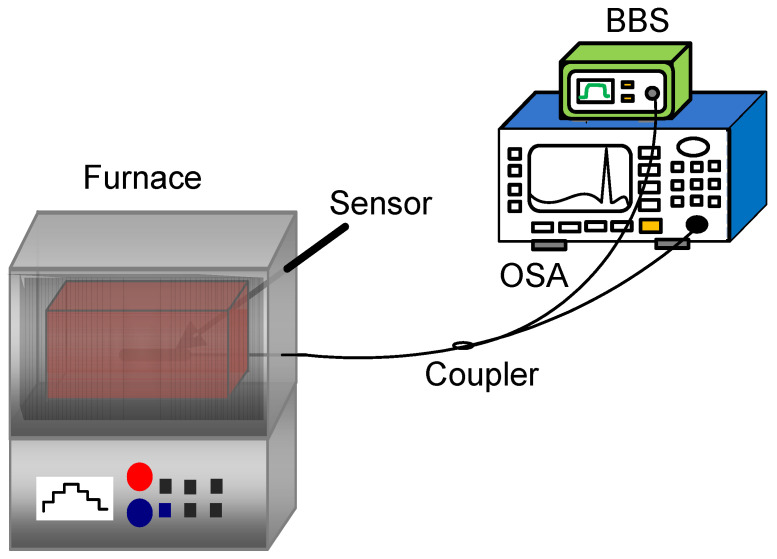
A schematic diagram of the experimental device for temperature measurement.

**Figure 13 materials-14-01028-f013:**
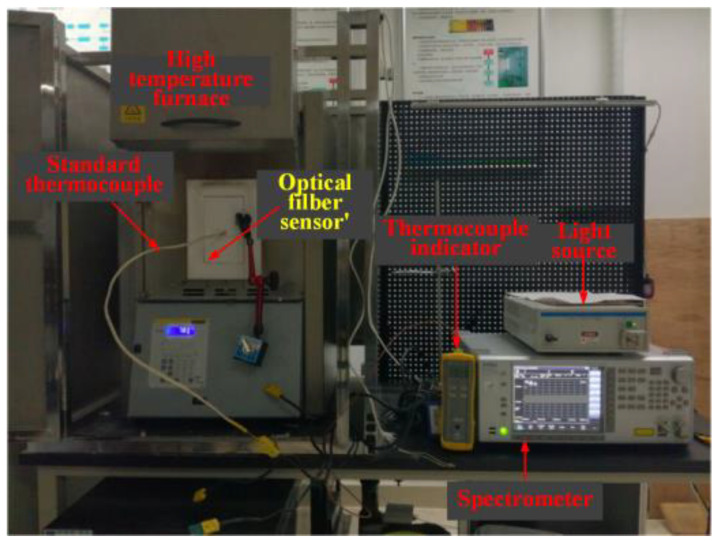
High temperature measurement experiment in laboratory.

**Figure 14 materials-14-01028-f014:**
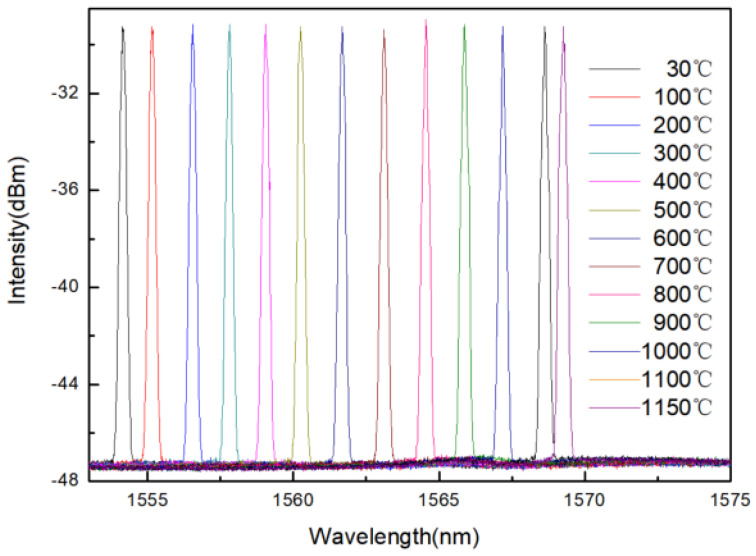
The central wavelength of the sensor drifts to the long wavelength with the increase of temperature.

**Figure 15 materials-14-01028-f015:**
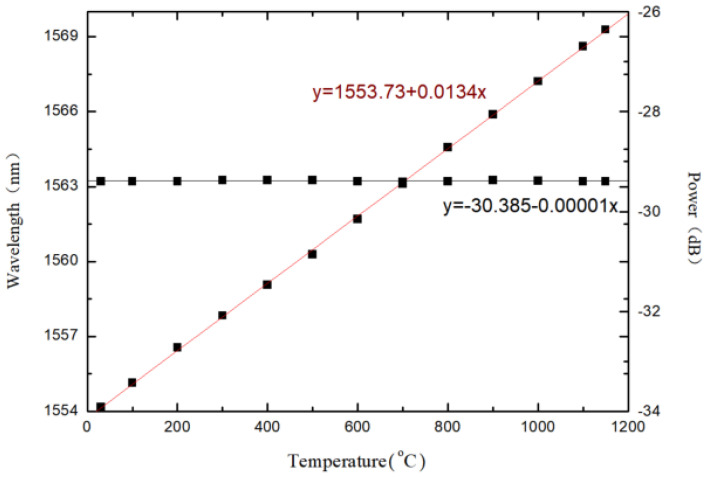
Temperature response sensitivity of the sensor.

**Figure 16 materials-14-01028-f016:**
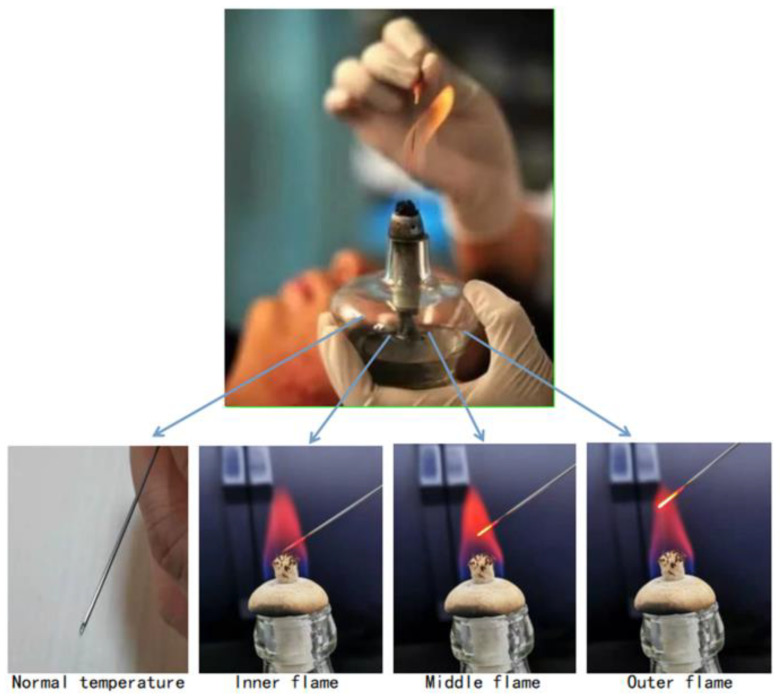
Needle burning experiment at different environments.

**Figure 17 materials-14-01028-f017:**
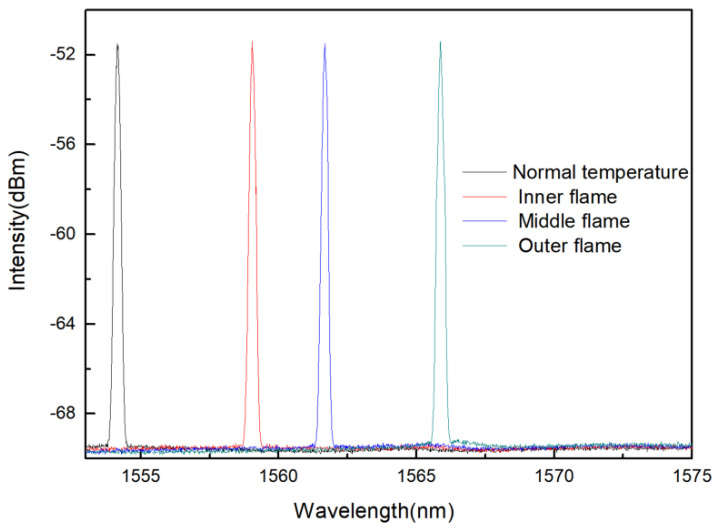
Reflectance spectrum in different environments.

**Figure 18 materials-14-01028-f018:**
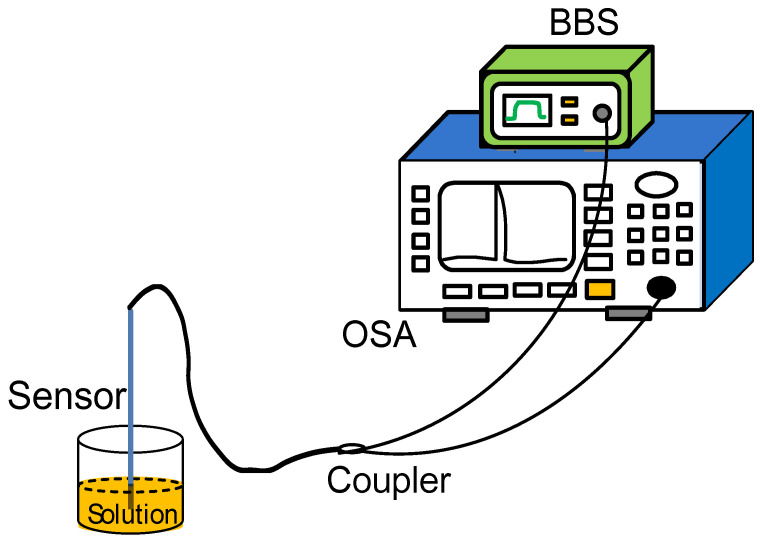
A schematic diagram of the experimental device for refractive index measurement.

**Figure 19 materials-14-01028-f019:**
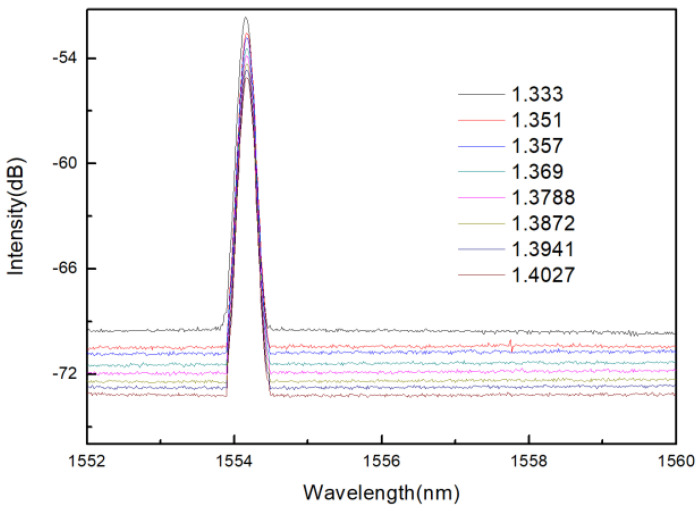
Reflection spectra of corroded FBG sensor with different refractive index.

**Figure 20 materials-14-01028-f020:**
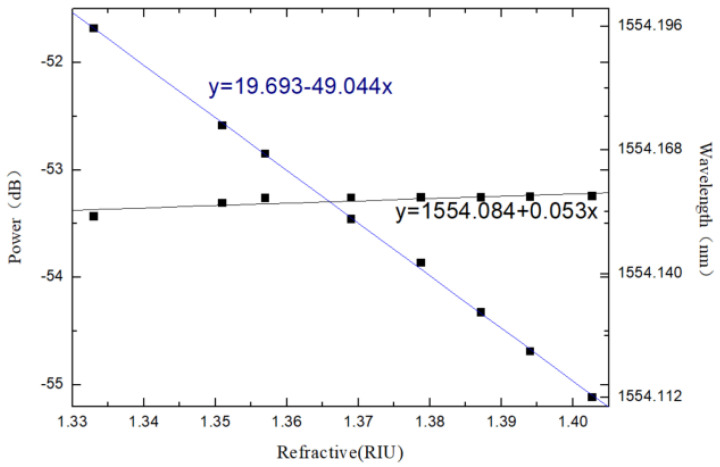
Refractive index response sensitivity of the sensor.

**Figure 21 materials-14-01028-f021:**
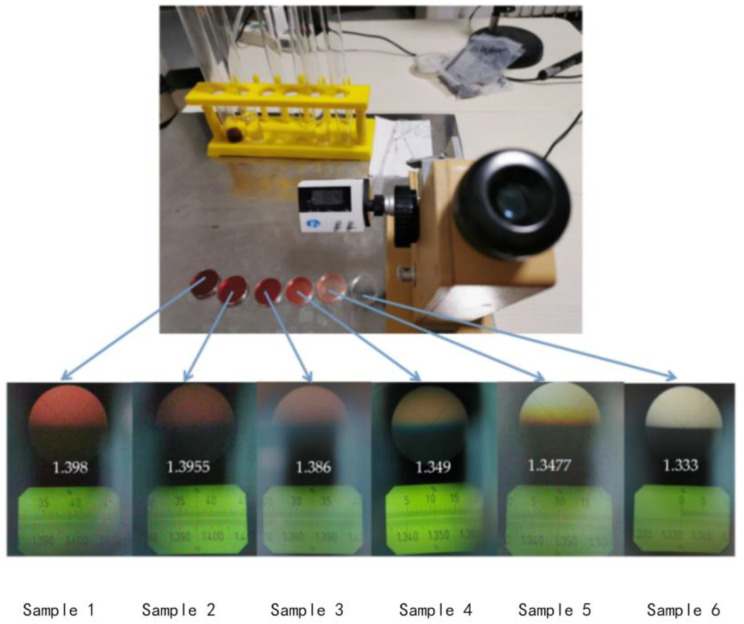
Refractive index experiment of blood samples with different refractive indices.

**Figure 22 materials-14-01028-f022:**
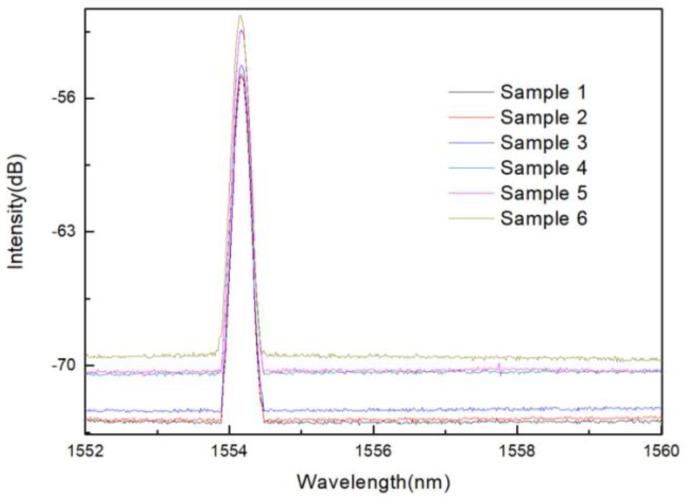
Spectra of blood samples with different refractive indices.

**Figure 23 materials-14-01028-f023:**
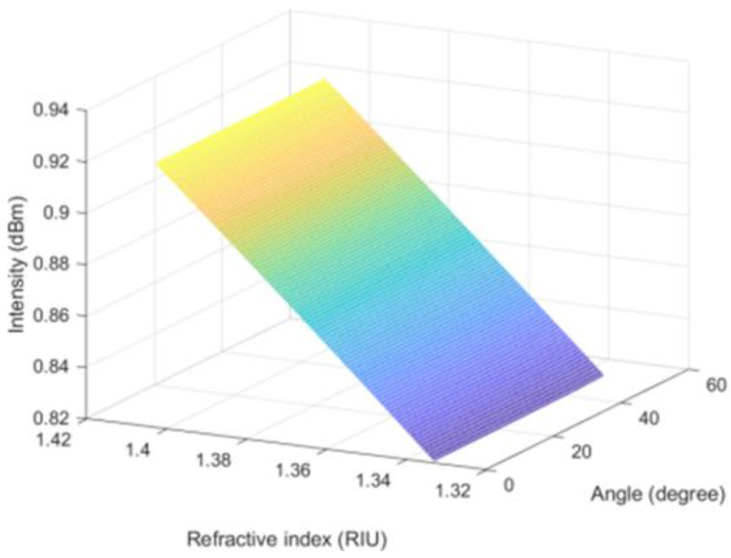
The intensity varying with the external refractive index under different angles.

**Figure 24 materials-14-01028-f024:**
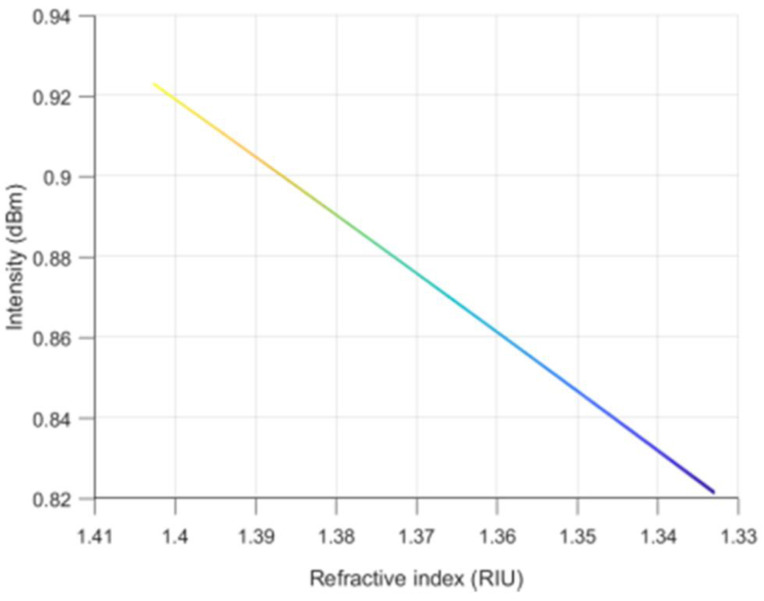
The intensity varying with external refractive index.

**Figure 25 materials-14-01028-f025:**
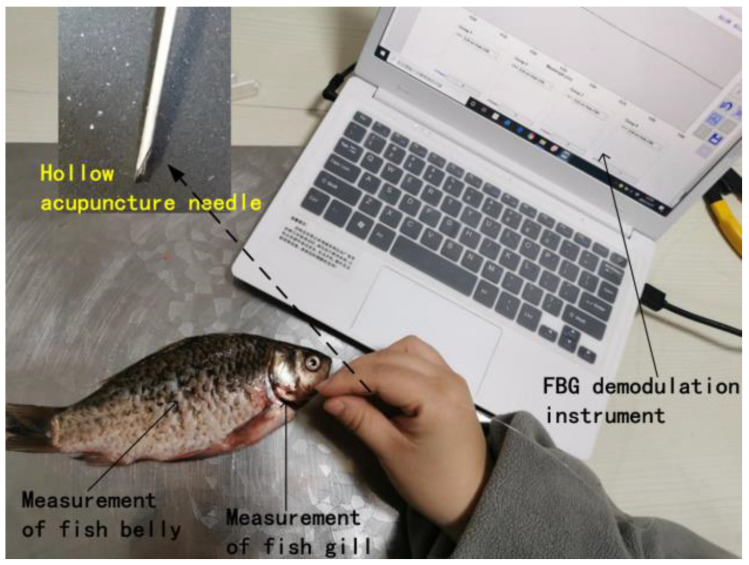
Experiment site of acupuncture on different parts of fish.

**Figure 26 materials-14-01028-f026:**
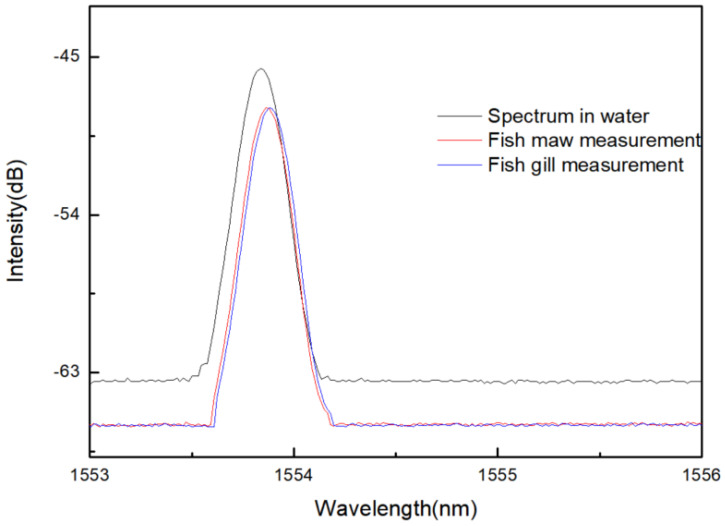
Experimental spectrum during acupuncture.

**Table 1 materials-14-01028-t001:** Production conditions and results of different FBGs.

FBG Type	Materials	Hydrogen Carrier	Anneal	Modulation Mode	Reflectivity	Temperature
Common FBG	Glass fiber	Yes	No	Refractive index modulation	80%	<400 °C
Regenerative FBG	Glass fiber	Yes	No	Refractive index modulation	20%	<1000 °C
Femtosecond pure silica FBG	Pure silica fiber	No	Yes	Physical destruction	70%	>1000 °C

**Table 2 materials-14-01028-t002:** Calibration result of temperature measurement deviation.

Sensor Number	Mach Number [Ma]	Reference Temperature [t_g0_/°C]	Temperature to be Calibrated [t_i_/°C]	Temperature Difference [Δt_i_/°C]	Extended Uncertainty [U(Δt_i_)/°C (k = 2)]
Xjtuipe-lbsfs	0.198	890.2	889.7	0.5	17
	0.202	1100.8	1100.3	0.5	19

**Table 3 materials-14-01028-t003:** Measure the temperature at different environments through spectral intensity.

Measuring Position	Normal Temperature	Inner Flame	Middle Flame	Outer Flame
Wavelength (nm)	1554.14	1559.52	1562.64	1565.56
Temperature (°C)	30.6	432.1	664.9	882.8

**Table 4 materials-14-01028-t004:** Measure the refractive index of each sample through spectral intensity.

Liquid Sample	1	2	3	4	5	6
Intensity (dBm)	−48.870	−48.747	−48.281	−46.467	−46.403	−45.682
Measured refractive index (RIU)	1.398	1.3955	1.386	1.349	1.3477	1.333

**Table 5 materials-14-01028-t005:** Temperature and refractive index measurement values at different measurement points in acupuncture.

Measuring Position	Spectrum in Water	Fish Maw Measurement	Fish Gill Measurement
Wavelength (nm)	1553.84	1553.86	1553.90
Intensity (dBm)	−45.682	−47.938	−48.331
Temperature (°C)	8	10	12
refractive index (RIU)	1.333	1.379	1.387

## Data Availability

Data sharing is not applicable to this article.

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
