# Peer review of "Simultaneous Measurement of Temperature and Refractive Index Using High Temperature Resistant Pure Quartz Grating Based on Femtosecond Laser and HF Etching"

_materials, 2021, doi:10.3390/ma14041028_

Round 1

Reviewer 1 Report

The authors presents fiber optic sensor for Simultaneous measurement of temperature and refractive index using high temperature resistant pure quartz grating based on femtosecond laser and HF etching. This type of fiberoptic sensor has already been studied and presented by other researchers in the field. I have some major concerns regrading the fabricated sensor' working principle and its characterization process. The sensor was glued to a stainless steel needle. This means different materials with different thermal coefficient of expansion were glued together. Once the temperature is raised, they will expand with different rates and as a result tension will be applied to the fiber optic sensor. We know that the FBG would be sensitive to the strain and thus it could interduce some red or blue shift on the spectrum. Also, I am not sure why the RI sensitivity in the range of 1.333 to 1.4027 was found to be linear. All in all, I don't recommend this paper to be published in Materials Journal as is.

Author Response

Dear Reviewer: 
    We have learned much from the your comments, which are fair and constructive. After carefully studying the comments and your advice, we have made corresponding changes. And we also add some experiments in this manuscript. 

   The attachment is the answers and revisions we have made in response item by item. Thank you for downloading and reviewing.

Reviewer 2 Report

Dear colleagues, thank you for your article and interesting diving in acupuncture from the point of its burning character. But, I have some questions and some large doubts.

  1. Line 25. The term "etching" is used more often than "corrosion"
  2. Line 28. The term "modulation" is hardly applicable in this context, since it implies the presence of a stable carrier, and more refers to information systems, rather than measurement systems.
  3. Line 29. Not clear. What "several peaks reflected from the grating" are we talking about?
  4. Table 1. It is not clear what “ordinary fiber” or “ordinary grating” means. 42 years have passed since the birth of the fiber Bragg grating, thousands of varieties of gratings have been created, which one is considered as “ordinary”? There are "ordinary" gratings on sapphire, which also measure 1200 ° C.
  5. Line 99-100. A very controversial statement. Doesn't stand up to criticism. If the authors want to measure the loss in a fiber caused by the leakage of core modes through a cladding with variable parameters, it is necessary to correctly and clearly substantiate the advantages of this method over the method with wavelength measurement.
  6. Section 2.1 indicates the material differences for the three types of gratings. In this case, the most detailed information is about a pure silicon fiber, but it is said that optical fiber cladding is doped with something. What is the optical fiber cladding doped with? Thus, I refer to the remarks on item 4
  7. Line 173. It is not clear what “effective refraction” is and what variables in (2) mean what. There is a feeling that the authors are careless about the basic equation that describes the operation of their sensor.
  8. Fig. 4 If the figure is to be believed, as it is drawn, there should be a strong Fresnel reflection at the end of the fiber, not any end loss shown there. If so, then it is not clear whether Fig. 4 is true and whether fig. 5 Either the authors missed something when designing the pic. 4 In this case, the characteristics of the grating will play a huge role, and we do not observe them in the article. The heading for section 3.2 deals with fiber termination, but the section itself does not.
  9. Next are the well-known sections on recording and etching fibers, which, if they have nothing new in them, are of no interest to anyone.
  10. Section 4.1. Temperature Measurements demonstrates the ability of the manufactured sensor to measure temperature. However, this has been proven by 40 years of experience using gratings in temperature measurements. The process of changing the temperature of the sensor at the acupuncture point is not shown. Therefore, all the previous scheduling of acupuncture theory and the purpose of the article is meaningless.
  11. Likewise, a great deal of work has been devoted to measuring glucose concentration. Doubt is the stability of the lattice wavelength to concentration changes of almost 0.08 RIU (see equation (2)).
  12. Section 5 presents a general theory of two-parameter measurements, which, unfortunately, also does not correlate with the theory of acupuncture. In addition, it is unlikely that at a temperature of 1000 or even 100 ° C, any physiological fluid will be present at the acupuncture point, for which it will be necessary to measure the refractive index.

Author Response

Dear Reviewer:
      We have learned much from the your comments, which are fair, encouraging and constructive. After carefully studying the comments and your advice, we have made corresponding changes. And we also add some experiments in this manuscript.

      The attachment is the answers and revisions we have made in response item by item. Thank you for downloading and reviewing.

Reviewer 3 Report

LIne 25.

The term "etching" is used for FBG sensors instead of "corrosion".

Line 29.

How reflection from a single FBG can generate "some reflection peaks"? The author's comments are required. 

Line 49.

The next sentence does not understandable: "If it is not red, it will not cure the disease and harm people."

Line 55.

The next sentence does not understandable: "After the fire needle enters the body, the temperature is difficult to maintain a stable temperature in the body."

Line 88. Table 1. 

The color (laser wavelength) and method (light generation method) are compared in Table 1 in the column "Laser type". It is unacceptable.

Line 122. 

It is a controversial statement. The intensity of FBG reflection is susceptible to a number of effects. The central wavelength is changed with the refractive index changing, look to Equation (2) in line 175.

Line 132. Figure 1.

1. The labels in Figure 1 is unreadble. 

2. It is understandable the abscissa axis. The title  "20 (deg.)" is used, while labels 20, 40, 60, and 80 are presented at the axis. 

3. The used temperature scale (Celsius, Kalvin, or even Fahrenheit) was not denoted.

Lines 172, 175. Equations (1) and (2).

Equations (1) and (2) are not accompanied by definitions of variables.

Lines 182-184. 

The intensity of the reflection coefficient can not be used as a base of measurements common.

Line 189. Figure 4. The strong Fresnel reflection should be at the end of the fiber. The author's comments are required.

The section "3. Production and packaging" does not contain novelty.

Lines 199-289. Section 3. 

The section "3. Production and packaging" does not contain novelty. The commonly used methods for BBG writing, fiber etching, and fixing of a sensor in a framework are described only. Is it interesting for readers really?

Lines 299-301.

How can authors provide the resolution of 1℃ in an optical spectrum analyzer with wavelength resolution: 0.02nm? Because the wavelength shift of FBG is 10 pm per 1℃. The author's comments are required.

Line 302.

Figure 11 contains the labels in china language. 

Lines 306-312. 

The linear dependence of FBG wavelength on temperature is doubtful. Our experiments show the square dependence of FBG wavelength on temperature. The author's comments are required.

Lines 323-351.

The section "4.2 Refractive index measurement" is doubtful. Figure 15 shows the intensity of reflection spectra on the different refractive index only. The central wavelength of FBG will be shifted on also, the different refractive index, as I mind.

Line 359. 

The ξ and α are absent in Equation (6). 

****

Taking into account the comments, I think that the article requires major revision.

Author Response

(The authors gave the same response as above.)

Round 2

Reviewer 1 Report

I have expected to see clear answers to my comments.

Author Response

Responses to Reviewer’s Comments

        Thank you very much for your comments about our paper submitted to Simultaneous measurement of temperature and refractive index using high temperature resistant pure quartz grating based on femtosecond laser and HF etching ( Materials; 1074328). We have learned much from your comments and we have made corresponding changes. In the manuscript, the green highlight is the second modification, and the gray highlight is the mark of the first modification.

        The attachment is the answers and revisions we have made in response to the reviewer's questions and suggestions item by item. Thank you for checking, and in the new year, we wish you success in your scientific research and all your wishes come true.

   With best regards!

                                                                                                    Yours faithfully

Reviewer 2 Report

Line 103-104. The inserted text does not explain the answer to my comment. The text is very controversial, it doesn't stand up to criticism. If the authors want to measure the loss in a fiber caused by the leakage of core modes through a cladding with variable parameters, it is necessary to correctly and clearly substantiate the advantages of this method over the method with wavelength measurement.

The process of changing the temperature of the sensor at the acupuncture point is not shown. Therefore, all the previous scheduling of acupuncture theory and the purpose of the article is meaningless.
Doubt is the stability of the lattice wavelength to concentration changes of almost 0.08 RIU (see equation (2)).

There is the mistake in Eq(2). Last term in Eq(2) must have a ΔT multiplyer. Moreover, the Eq(2) looks uncorrect, please compare with:

ΔλB=2(Λ∂n/∂l+n∂Λ/∂ll+2(Λ∂n/∂T+n∂Λ/∂TT

The authors comments are requaried.

Section 5 presents a general theory of two-parameter measurements, which does not correlate with the theory of acupuncture.

I think that the authors need to take these comments into account in the article. The manuscript can be published after major revisions.

Author Response

(The authors gave the same response as above.)

Reviewer 3 Report

If the "author's comments are required" it is mentioned that the author's comments are really required.

I am forced to keep the following remarks:

Line 49.
The next sentence does not understandable: "If it is not red, it will not cure the disease and harm people."
A reference to the original publication is required.

Line 55.
I suggest using another formulation:
"After the fire needle enters the body, it is difficult to maintain a stable temperature in the body"
instead of
"After the fire needle enters the body, the temperature is difficult to maintain a stable temperature in the body".

Line 136. Figure 1.
1. The labels in Figure 1 is unreadable.
2. The label of the abscissa axis is not explained in the text.
3. The used temperature scale (Celsius, Kalvin, or even Fahrenheit) of the abscissa axis was not denoted.

Lines 141, 142, 150,
The used temperature scale (Celsius, Kalvin, or even Fahrenheit) was not denoted.

Line 178-179.
The term "grating pitch" is used for FBG instead of "grating period".

Line 187
The term "wavelength drift" is used for FBG instead of "wavelength shift"

Lines 177, 181. Equations (1) and (2).

Equations (1) and (2) are not accompanied by definitions of used variables.

Lines 182-184.

The intensity of the reflection coefficient can not be used as a base of measurements common.

Line 189. Figure 4. The strong Fresnel reflection should be at the end of the fiber. The influence of Fresnel reflection does not investigate. The simulation of Fresnel reflection is not explained. The author's comments are required.

****

Taking into account the comments, I think that the article requires minor revision.

Author Response

(The authors gave the same response as above.)

Round 3

Reviewer 1 Report

I still don't believe this research has the required "originality", "significance of content", "quality of presentation", and "scientific soundness". Therefore, I don't recommend its publication.  

Author Response

Dear Reviewer: 
     Thank you very much for your comment about our paper submitted to Simultaneous measurement of temperature and refractive index using high temperature resistant pure quartz grating based on femtosecond laser and HF etching ( Materials; 1074328). We have learned much from your comments and we have made corresponding changes. In the manuscript, the blue highlight is the latest modification, and the gray highlight is the mark of the previous modification.

       The attachment is the revision we have made in response. Thank you for downloading and reviewing. 

Reviewer 2 Report

There is the mistake in Eq(2). Last term in Eq(2) must have a ΔT multiplyer. Moreover, the Eq(2) looks uncorrect, please compare with:

ΔλB=2(Λ∂n/∂l+n∂Λ/∂ll+2(Λ∂n/∂T+n∂Λ/∂TT

The authors comments are requaried.

I think that the authors need to take these comments into account in the article. The manuscript can be published after minor revisions.

Author Response

(The authors gave the same response as above.)
